# Natural Polyphenols May Normalize Hypochlorous Acid-Evoked Hemostatic Abnormalities in Human Blood

**DOI:** 10.3390/antiox11040779

**Published:** 2022-04-14

**Authors:** Tomasz Misztal, Agata Golaszewska, Natalia Marcińczyk, Maria Tomasiak-Łozowska, Małgorzata Szymanowska, Ewa Chabielska, Tomasz Rusak

**Affiliations:** 1Department of Physical Chemistry, Medical University of Białystok, 15089 Białystok, Poland; g.szymanowska@o2.pl (M.S.); tomasz.rusak@umb.edu.pl (T.R.); 2Department of General and Experimental Pathology, Medical University of Białystok, 15089 Białystok, Poland; agata.golaszewska@umb.edu.pl; 3Department of Biopharmacy, Medical University of Białystok, 15089 Białystok, Poland; natalia.marcinczyk@umb.edu.pl (N.M.); ewa.chabielska@umb.edu.pl (E.C.); 4Department of Allergy and Internal Diseases, Medical University of Białystok, 15276 Białystok, Poland; maria.tomasiak-lozowska@umb.edu.pl

**Keywords:** platelets, aggregation, thrombus formation, fibrinolysis, antioxidants, hypochlorous acid, myeloperoxidase, quercetin, epigallocatechin gallate, resveratrol

## Abstract

During pathogen invasion, activated neutrophils secrete myeloperoxidase (MPO), which generates high local concentrations of hypochlorous acid (HOCl), a strong antimicrobial agent. Prolonged or uncontrolled HOCl production may, however, affect hemostasis, manifesting in inhibition of platelet aggregation and thrombus formation and in elevated fibrin density and attenuated fibrinolysis. In this report, we investigated whether three plant-derived polyphenols with well-known antioxidant properties, i.e., quercetin (Que), epigallocatechin gallate (EGCG), and resveratrol (Resv), at concentrations not affecting platelet responses per se, may normalize particular aspects of hemostasis disturbed by HOCl. Specifically, Que (5–25 μM) and EGCG (10–25 μM) abolished HOCl-evoked inhibition of platelet aggregation (assessed by an optical method), while the simultaneous incubation of platelet-rich plasma with Resv (10–25 μM) enhanced the inhibitory effect of HOCl. A similar effect was observed in the case of thrombus formation under flow conditions, evaluated in whole blood by confocal microscope. When plasma samples were incubated with HOCl, a notably higher density of fibrin (recorded by confocal microscope) was detected, an effect that was efficiently normalized by Que (5–25 μM), EGCG (10–25 μM), and Resv (5–25 μM) and which corresponded with the normalization of the HOCl-evoked prolongation of fibrinolysis, measured in plasma by a turbidimetric method. In conclusion, this report indicates that supplementation with Que and EGCG may be helpful in the normalization of hemostatic abnormalities during inflammatory states associated with elevated HOCl production, while the presence of Resv enhances the inhibitory action of HOCl towards platelets.

## 1. Introduction

During pathogen invasion, human neutrophils generate large quantities of hypochlorous acid (HOCl), a strong oxidant and chlorinating agent, efficient in killing bacteria and pathogenic fungi [1,2,3]. HOCl is produced from hydrogen peroxide and a chloride ion by the heme-containing enzyme myeloperoxidase (MPO) localized in neutrophils’ azurophilic granules and released upon activation both to phagocytic vesicles (phagosomes) and into the extracellular space (plasma or interstitial matrix) [3,4]. MPO is also an abundant constituent of neutrophil extracellular traps (NETs), unique extracellular structures composed of uncondensed chromatin decorated with nuclear and cytosolic proteins with antimicrobial activity, formed by neutrophils in response to the presence of pathogens [5,6]. Different research groups estimated that up to 80% of the H_2_O_2_ generated by activated neutrophils during “respiratory burst” are used to form HOCl in a range from 20 µM/h [7] to 400 µM/h of chlorinating agent [8]. At the inflammation site, the local HOCl level can reach millimolar concentrations [9]. A chronic, acute, or malcontrolled inflammatory state may lead to overproduction of HOCl, resulting in chemical modification of plasma proteins, including those involved in hemostasis [10,11,12,13,14], and inhibition of platelet responses [15,16]. Our recent observations made in an in vitro system indicates that clinically relevant (micromolar) concentrations of HOCl inhibit platelet aggregation, secretion, and thrombus formation and that this MPO’s product alters fibrin clot structure (towards higher density) and attenuates fibrinolysis in human blood [16]. This suggests that HOCl may contribute to a thrombotic (by making a clot more resistant to lysis) or a bleeding phenotype (via inhibition of platelets). An example of clinical condition where elevated HOCl production coincides with decreased platelet contractility (and thus with inhibited clot retraction rate) and with attenuated fibrinolysis is asthma [17,18].

Since elevated plasma MPO has been established as a risk factor associated with cardiovascular diseases [19,20], searching for new strategies to normalize hemostatic responses in the presence of HOCl is of clinical significance.

In this report, we investigated whether three plant-derived polyphenolic compounds with widely recognized antioxidant properties, i.e., quercetin (3,3′,4′,5,7-pentahydroxy flavone; Que) [21,22], epigallocatechin gallate (EGCG) [22,23], and resveratrol (3,5,4′-trihydroxy-trans-stilbene; Resv) [22,24,25], are capable prevent HOCl-evoked hemostatic abnormalities in human blood. Next to their free radical-scavenging features, all these three compounds have been identified as anti-platelet agents, reducing platelet adhesion, secretion, and aggregation in vitro at relatively high (micromolar) concentrations (see references below). The biochemical mechanisms of this anti-platelet action include: inhibition of cyclooxygenase and thromboxane A_2_ formation [26], reduction of phospholipase Cβ and protein kinase C activity, and flattening of store-operated calcium entry [27], as well as stimulation of the NO/cGMP pathway [28] (Resv), inhibition of p38 MAPK and ERK-1 and -2 kinases [29] (EGCG), inhibition of phospholipase Cγ2 and tyrosine kinases Syk and Fyn [30], decrease of phosphoinositide-3-kinase and MAP kinases activity, and stimulation of cAMP production and VASP phosphorylation [31] (Que).

However, the idea that lower concentrations of the above antioxidants (incapable of affecting hemostasis per se) may normalize hemostatic disturbances evoked by naturally occurring strong oxidants, e.g., HOCl, has never been evaluated. This issue is crucial since global antioxidant supplementation is increasing and an antioxidant-rich diet is widely recognized as beneficial for the prevention of diseases, including cardiovascular and inflammation-related ones [32,33].

To test this hypothesis, we established anti-platelet concentration ranges for all three polyphenolic compounds in our experimental conditions and further used concentrations below the anti-platelet range to study their effect on HOCl-inhibited platelet aggregation, thrombus formation under flow, and fibrin clot structure and fibrinolysis in human blood in vitro.

## 2. Materials and Methods

### 2.1. Reagents

In this study, we used NaOCl solution as a “hypochlorous acid” (pKa 7.45). This refers to ~50% ionized mixture of HOCl and ClO^−^, existing at physiological pH (as “HOCl” throughout the text). Before each experiment, the exact concentration of HOCl was assessed by spectrophotometric measurement using a 292 nm wavelength and a molar extinction coefficient of 350 M^−1^cm^−1^. Quercetin, resveratrol, epigallocatechin gallate, and sodium hypochlorite solution were from Sigma-Aldrich (Merck). Collagen (type I) was from Chrono-Log Corp. Alexa Fluor 488-conjugated human fibrinogen was from Thermo Fisher Scientific. Alteplase (recombinant human tissue plasminogen activator) was from Boehringer Ingelheim.

### 2.2. Blood Collection and Preparation

Venous blood was collected from healthy volunteers with minimum trauma and stasis via a 21-gauge needle (0.8–40 mm) onto 130 mM trisodium citrate. All procedures were conducted in accordance with the principles of the Declaration of Helsinki and the study was approved by the local Ethics Committee on human research (permission number APK.002.96.2020). Platelet-rich plasma (PRP) was obtained by centrifugation of whole blood at 200× *g* for 20 min. Platelet-poor plasma was obtained by centrifugation of whole blood at 2800× *g* for 10 min.

### 2.3. Platelet Aggregation Measurements

Aggregation of blood platelets was assessed in a freshly prepared platelet-rich plasma. An optical aggregometer (Elvi Logos 840; Milan, Italy), connected with a computer via e-corder 401 (eDAQ Pty Ltd.; Denistone East, Australia) was used. The increase in light transmittance through the stirred PRP samples—as a result of platelet aggregation [34]—was recorded using eDaq Chart software.

### 2.4. Thrombus Formation under Flow Study

Collagen (type I, 50 mg/mL)-coated, degreased coverslips were mounted onto a transparent, polycarbonate-made parallel-plate flow chamber (50 μm in depth, 3 mm in width, and 30 mm in length) [35] and pre-rinsed with Hepes buffer (pH 7.45) containing 0.1% BSA. Freshly collected, trisodium citrate-anticoagulated whole blood was immediately supplemented with PPACK (a direct thrombin and factor Xa inhibitor, 40 μM final concentration), divided into separate samples, and incubated with appropriate vehiculum (ethanol or water) or with HOCl or with indicated concentrations of the studied compounds, added 1 min before HOCl. Next, the samples were supplemented with DiO (1 μM, for 1 min, to stain platelets), after which they were supplemented with MgCl_2_ and CaCl_2_ solutions (to the final concentrations of 3 mM in both cases) and immediately perfused through the flow chamber for 4 min at a shear rate of 1000 s^−1^. End-stage fluorescence images of the thrombi formed on collagen were generated using a confocal laser scanning microscope [36]. The surface coverage by adhered platelets in the area of 4800 μm^2^ was calculated with the use of free software (Rasband, 1997–2016 [37]).

### 2.5. Confocal Analyses of Clot Structure

Platelet-poor plasma samples were preincubated at 37 °C for 5 min with appropriate vehiculum (ethanol or water) or with HOCl or with indicated concentrations of studied compounds, added 1 min before HOCl. After the incubation period, AF488-labeled human fibrinogen was added to all samples (to a final concentration of 60 μg/mL) and the clotting was triggered by the addition of CaCl_2_ to a final concentration of 20 mM. After 2 h incubation in a dark, humid chamber (at 37 °C), the fibrin clots were analyzed with the use of the confocal microscope towards relative fibrin density. The preparation and visualization of platelet–fibrin clots is essentially described in [38]. The samples were examined after the transfer to microchamber slides (Ibidi μ-slide VI; Animalab). At least 10 pictures of different spots of each clot were recorded. Relative clot density was considered as the number of fibrin fibers crossing a single, randomly placed 30 μm-long straight line.

### 2.6. Fibrinolysis Assessment in Plasma

The kinetics of fibrin clots lysis in plasma samples was evaluated by the turbidimetric method described in detail in [39]. Plasma samples were preincubated at 37 °C for 5 min with appropriate vehiculum (ethanol or water) or with HOCl or with indicated concentrations of studied compounds, added 1 min before HOCl. After incubation, coagulation was triggered by the addition of CaCl_2_ (to 20 mM final concentration). The time to reach 50% and 100% of lysis was measured as the time necessary for a decrease in absorbance after reaching the maximal turbidity in the presence of the tissue plasminogen activator (Alteplase, 200 ng/mL final conc.).

### 2.7. Determination of Thiol Groups in HOCl-Exposed Plasma

Thiol content in plasma samples exposed to HOCl was measured using the Ellman reaction. Sulfhydryl group levels were calculated by using the molar absorption coefficient for DTNB (5,5′-dithiobis-(2-nitrobenzoic acid)) at 37 °C (DTNB ε_412_ = 13,800 M^−1^cm^−1^) [40].

### 2.8. Data Analyses

GraphPad Prism 5 (GraphPad Software, San Diego, CA, USA) was utilized to analyze the obtained data. The differences between two groups were assessed by means of the Mann–Whitney U test. The data are presented as the means ± S.D. of the number of determinations (n) or in a “percent of control” fashion. In all cases, a *p*-value < 0.05 was considered to be significant.

## 3. Results

During preliminary experiments, we defined HOCl concentrations producing ~45–55% inhibition (or increase, in case of clot density examination) in regard to the studied response. Such selected concentrations of HOCl were used in particular research tasks, i.e., a platelet aggregation study, an evaluation of thrombus formation under flow, a quantification of clot density, and an assessment of clot lysis rate.

### 3.1. Concentration-Dependent Effects of Que, EGCG, and Resv on Platelet Aggregation

Que, EGCG, and Resv reduced collagen-evoked aggregation of human platelets (in PRP) in a dose-dependent manner (Figure 1A). Established IC_50_ values were: 60 ± 5 μM for Que, 100 ± 8 for EGCG, and 50 ± 6 for Resv. When PRP samples were incubated with 200 μM HOCl, about 50% inhibition of aggregation was recorded. A 1 min long preincubation with Que or EGCG (before adding HOCl) normalized platelet aggregability (Figure 1B). This effect started to be significant from 5 μM (for Que) and 10 μM (for EGCG) concentrations. Conversely, Resv was unable to normalize HOCl-evoked inhibition of platelet aggregability. The reduction of aggregation was deepened at studied concentration range of Resv (1–25 μM).

### 3.2. Effects of Que, EGCG, and Resv on Thrombus Formation under Flow

Que, EGCG, and Resv attenuated the formation of platelet thrombi on collagen-coated surfaces under flow (arterial shear conditions, i.e., 1000 s^−1^, Figure 2 upper panel). Significant inhibition was observed at 50 μM (for Que and Resv) and at 200 μM (for EGCG). After incubation of whole blood samples with HOCl (500 μM), about a 50% reduction of thrombus formation was observed. This effect was prominent and comparable to the effect of reference antiplatelet agents—aspirin and iloprost [41,42]. Preincubation (1 min) with Que (10–50 μM) or EGCG (25–100 μM) normalized HOCl-evoked diminishing of thrombus formation (Figure 2 lower panel). Resv, at a concentration range of 25–100 μM, was unable to normalize HOCl-evoked impairment of thrombus formation but a synergistic inhibition was observed (Figure 2 lower panel).

### 3.3. Normalization of HOCl-Evoked Densification of Fibrin Clots by Que, EGCG, and Resv

Incubation of plasma samples with HOCl (250 μM) resulted in the formation of denser (about 40–50%) clots after recalcification. Preincubation (1 min) of samples with Que (5–25 μM), EGCG (10–25 μM), or Resv (5–25 μM) protected from HOCl-mediated fibrin densification (Figure 3). Que, EGCG, and Resv per se did not modulate fibrin density at the studied concentrations (data not shown).

### 3.4. Normalization of HOCl-Associated Impairment of Fibrinolysis by Que, EGCG, and Resv

Incubation of plasma samples with HOCl (125 μM) resulted in the formation of lysis-resistant clots after recalcification, i.e. prolongation required to obtain 50% and 100% of clots lysis was recorded. Que (10–50 μM), EGCG (10–50 μM), or Resv (5–50 μM) normalized fibrinolysis after 1 min preincubation (Figure 4). Que, EGCG, and Resv per se did not affect fibrinolysis at the studied concentrations (data not shown).

### 3.5. Counteraction against HOCl-Mediated Oxidations of Sulfhydryl Groups in Plasma by Que, EGCG, and Resv

Incubation of plasma samples with HOCl (125 μM) resulted in a substantial decrease in -SH group contents (Figure 5). This effect was normalized by the preincubation of plasma samples with Que, EGCG, or Resv (10–25 μM).

## 4. Discussion

In this report, we tested the hypothesis that plant-derived polyphenols—quercetin, epigallocatechin gallate, and resveratrol—might play a protective role against HOCl and thus normalize hemostatic responses disturbed by this stressor. This idea was based on the study of Hu et al., where it was shown that the addition of HOCl to native human plasma results in the consumption of albumin-associated sulfhydryl groups as well as hydroxyl group-bearing antioxidant (using ascorbic acid as an example) [43].

The results presented here show that each of the studied polyphenols exerts anti-HOCl action in human plasma. This was evidenced by observations that elevated fibrin density and the attenuated fibrinolysis associated with it—previously shown to be connected to HOCl-mediated oxidative modifications of fibrinogen [16]—were normalized in the presence of each of the studied compounds (Figure 3 and Figure 4). This was also supported by the observation that all three of the studied antioxidants offered protection from the HOCl-evoked decrease in the number of reduced -SH groups (markers of the oxidative action of this stressor) at concentrations corresponding to those normalizing clot density and fibrinolysis (Figure 5). The effects of these polyphenols on the HOCl-treated platelets are, however, more complex. Relatively high concentrations of Que, EGCG, and Resv reduced platelet aggregation and thrombus formation dose-dependently, whereas lower concentrations (below antiplatelet range) defended from HOCl-evoked inhibition of these platelet responses (Figure 1 and Figure 2), with the prominent exception of Resv. Used together, Resv and HOCl showed synergistic inhibition of platelet aggregation and thrombus formation under flow (Figure 1B and Figure 2 lower panel). One potential explanation is that HOCl and Resv may act on different—but complementary—signaling pathways within platelets with an additional inhibition as a net result. Resv has been shown to stimulate inhibitory pathways (e.g., the NO/cGMP pathway [28]) as well as to diminish calcium signaling [27] and activity of protein kinases essential for platelet activation [29]. Correspondingly, HOCl has been evidenced as an inhibitor of mitochondrial respiration and calcium influx in porcine platelets [15]. It is widely accepted that oscillating calcium increases in platelet cytosol are necessary for the conformational change of GPIIb/IIIa receptors (resulting in activation of these receptors) [44]—a prerequisite for stable platelet aggregation in the following manner: GPIIb/IIIa (on one platelet)–fibrinogen–GPIIb/IIIa (on the other platelet). Additionally, reduced calcium influx to platelets is expected to diminish thromboxane A_2_ (TxA_2_) formation and to reduce granule content release (secretion), since both processes are calcium-dependent [45,46]. The secretion of secondary activators (e.g., ADP) and the production of TxA_2_ are crucial for the activation of additional platelets and hence in aggregation progress and stable thrombus formation [41,42,47]. Such inhibited platelet aggregation may explain the attenuation of thrombus formation observed under flow conditions, where the size of formed aggregates was notably reduced (Figure 2). In the cases of Que and EGCG, we observed an anti-HOCl effect—in relation to platelet aggregation and thrombus formation—at concentrations corresponding to those protecting from HOCl impact in plasma. This suggests that the antioxidative properties are the primary mechanism of action of Que and EGCG at this concentration range (below the antiplatelet range).

The question arises whether supplementation with polyphenols during inflammatory states—in which the production of large quantities of HOCl are expected—would be worth considering, especially when a patient undergoes antiplatelet pharmacotherapy? One clinical trial suggests that EGCG could be safe when combined with classical antiplatelet agents, i.e., aspirin, clopidogrel, and ticagrelor [48]. Que and its metabolites have been found to enhance the antiplatelet effect of aspirin, with the indication that a substantial part of the investigation was performed using washed platelet suspension, i.e., an experimental model which is hard to extrapolate to in vivo situations [49]. One in vitro study showed no enhancement of platelet inhibition when Resv and clopidogrel were used together [50]. However, in another study, the authors showed that Resv may efficiently reduce collagen-evoked platelet aggregation in aspirin-resistant patients [51]. In aspirin responders, a combination of Resv with aspirin has been shown to synergistically reduce platelet aggregation [52]. Since aspirin is commonly used to reduce symptoms of inflammation (beyond its antiplatelet action) and polyphenols are increasingly recognized as beneficial antioxidants, the simultaneous exposure of certain individuals to aspirin and Resv during conditions associated with elevated HOCl production is probable. Such interactions may hypothetically lead to modulation of hemostatic response with consequences that are hard to predict. One may assume that ongoing pharmacotherapy and overall vascular homeostasis would additionally determine the clinical picture during the above-mentioned interactions. Therefore, there is an urgent need to explore interactions between polyphenols and classical antiplatelet drugs more deeply, considering broader concentrations of polyphenols (not only within the antiplatelet range). Our research can make a significant contribution to this field as a preliminary study.

Since widely used antiplatelet drugs, i.e., ADP receptor (P2Y12) antagonists (clopidogrel, ticlopidine, prasugrel), require bioactivation via cytochrome P450 (2C19 isoform) [53,54], potential interactions between polyphenols and CYP450 are a cause for concern. IC_50_ values were established in a range of micromolar concentrations for Que (11 µM –>30 µM) [55,56], EGCG (109 µM) [57], and Resv (11.6 µM–22.5 µM) [58,59]. However, since isolated microsomes or heterologously expressed CYP450 (2C19) were used in these studies, one may hypothesize that obtaining the effective inhibitory concentrations in specific tissues in vivo would require oral administration of notably higher concentrations of these polyphenols compared with those exerting anti-HOCl properties.

One limitation of this report is that the plasma concentration of polyphenols exerting anti-HOCl activity might be unreachable in vivo via oral administration. The bioavailability of polyphenols is a limiting factor since their concentration in plasma after taking polyphenol-rich tablets or drinking red wine (the richest source of resveratrol in the diet) is within the nanomolar range [60,61,62,63,64]. However, novel supplementation strategies, including the use of nanoparticles, liposomes, or dispersion techniques, are able to manage plasma concentrations of polyphenols at low micromolar ranges [65,66,67], making their protective role against HOCl probable. It has also to be stressed that our approach considered the specific range of the stressor concentrations, i.e., those producing about 50% of inhibition/augmentation (dependent on specific parameters). One can expect that lower concentrations of HOCl, also producing hemostatic abnormalities (and more likely to occur in the blood during inflammation [16]), would be neutralized by even lower—and hence more easily achieved in vivo—concentrations of specific antioxidants.

Since in this report platelet aggregation and thrombus formation were triggered solely by collagen, the question arises: What might be the impact of HOCl and antioxidants on platelet responses evoked by other platelet stimuli, e.g., ADP or thrombin receptor agonists? HOCl seems to be rather a weak inhibitor of thrombin receptor activating peptide (TRAP)-induced platelet activation, since a significant reduction of platelet secretion (evaluated as P-selectin exposure) triggered by this agonist was observed at a relatively high (≥0.5 mM) concentration of stressor [16]. On the other hand, 0.15 mM HOCl strongly inhibited platelet aggregation induced by ADP in a washed platelet suspension [68]. The potential impact of antioxidants on platelet responses, triggered by agonists other than collagen, in the presence of HOCl is a valid subject for future investigations.

It has to be stressed, also, that bolus addition of HOCl into a biological sample may not properly reflect the effect of a continuously produced stressor during inflammatory response, since the majority of added HOCl is likely to be entrapped in situ due to the highly reactive nature of this compound. On the other hand, it has been shown that constant production of relatively small amounts of reactive species (e.g., peroxynitrite) may exert at least a similar effect to the bolus addition of high concentrations of this stressor [69]. To mimic the in vivo situation more accurately, a purified MPO/H_2_O_2_/Cl^−^ system or activated neutrophils are preferable for future studies.

## 5. Conclusions

In this report, we present evidence that particular polyphenols may act in different manners in human blood—as negative regulators of platelet responses (at high concentrations) or as factors normalizing hemostasis and fibrinolysis in the presence of a naturally occurring stressor, i.e., HOCl (at lower concentrations). Quercetin and epigallocatechin gallate seem to exert promising anti-HOCl potential and hence may be helpful in the normalization of hemostatic/fibrinolytic abnormalities during inflammatory states associated with elevated HOCl production. Conversely, resveratrol normalizes HOCl action only in platelet-free plasma, showing synergistic platelet inhibition in the presence of HOCl.

It has to be emphasized that the future development of the above polyphenols is subject to several limitations. The precise mechanism of action of these polyphenols should be elucidated and more complex studies should be conducted, including studies using vivo models, in order to identify potential health benefits as well as threats resulting from their usage, especially during inflammatory states and ongoing pharmacotherapy. Additionally, to cover the required concentrations of certain polyphenols in plasma, the novel delivery strategies that will be desirable to use, will—at least initially—elevate the expense of therapies as well as restrict them to the clinical environment.

In a longer perspective, coupling of the monitored supplementation of certain polyphenols using novel delivery strategies with a clinical practice might prospectively serve as a new potential strategy to normalize hemostatic abnormalities during massive HOCl production (inflammatory states). Potentially, other reactive species with evidenced disruptive action on hemostasis, e.g., peroxynitrite [38,70,71], might also be neutralized by certain antioxidants. However, as in the case of HOCl, the precise mechanism of such interactions must be revealed.

## Figures and Tables

**Figure 1 antioxidants-11-00779-f001:**
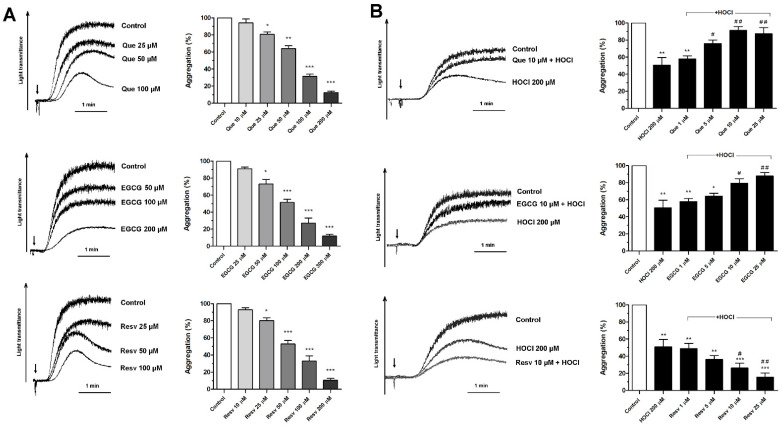
Effects of quercetin, epigallocatechin gallate, and resveratrol on platelet aggregation and on HOCl-evoked inhibition of aggregation. Samples of PRP were incubated with quercetin (Que), epigallocatechin gallate (EGCG), or resveratrol (Resv) for 5 min (**A**) followed by the addition of collagen (arrow, 5 μg/mL), and platelet aggregation, measured as light transmittance through platelet suspension, was recorded. (**B**) PRP samples were supplemented with HOCl (200 μM, for 5 min) or with indicated concentrations of Que, EGCG, or Resv added 1 min before HOCl. Next, aggregation was triggered by the addition of collagen (arrow, 5 μg/mL). Control samples contained an appropriate volume of vehiculum: ethanol (used as a control for Que and Resv) or water (used as a control for EGCG). Representative aggregation curves from one experiment (out of six) and extent of aggregation are presented. Aggregation obtained in the presence of only collagen was considered as maximal aggregation. Data are means ± S.D. from n = 6 experiments. * *p* < 0.05; ** *p* < 0.01; *** *p* < 0.001 vs. control (sample supplemented with collagen, (**A**,**B**)). # *p* < 0.05; ## *p* < 0.01 vs. sample supplemented with HOCl 200 μM (**B**).

**Figure 2 antioxidants-11-00779-f002:**
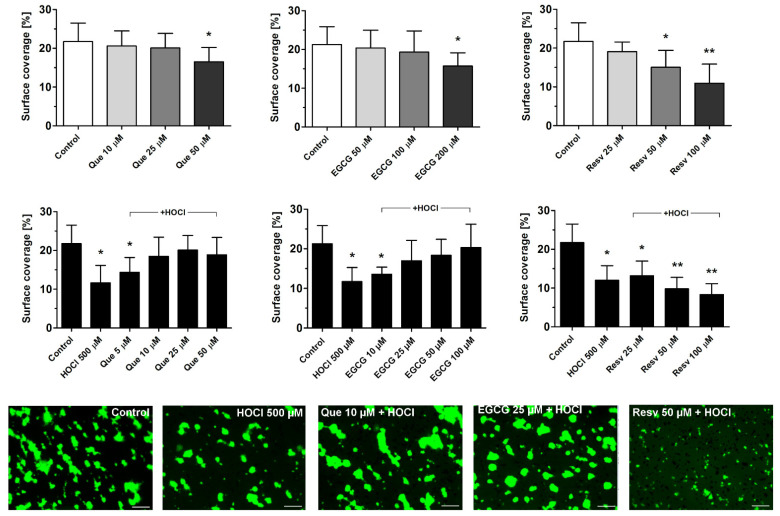
Effects of quercetin, epigallocatechin gallate, and resveratrol on thrombus formation under flow and on HOCl-mediated reduction of thrombus formation. PPACK-anticoagulated whole blood samples (supplemented with DiO to visualize platelets) were incubated with appropriate vehiculum (ethanol or water) or with indicated concentrations of Que, EGCG, or Resv (for 5 min, **upper panel**). Next, samples were perfused over collagen-coated surfaces at a shear rate of 1000 s^−1^ to form thrombi. Surface coverage area was calculated from end-stage confocal pictures. **Lower panel**: samples were incubated with HOCl (500 μM, for 5 min) or with Que, EGCG, or Resv added 1 min before HOCl and thrombus formation was performed under conditions as in **upper panel**. Representative thrombi obtained in the presence of HOCl or combinations of HOCl with selected concentrations of Que, EGCG, or Resv are presented in **lower panel**. Data are means ± S.D. from n = 3 experiments. * *p* < 0.05; ** *p* < 0.01 vs. control. Distance bar is 10 μm. Flow direction was from left to right.

**Figure 3 antioxidants-11-00779-f003:**
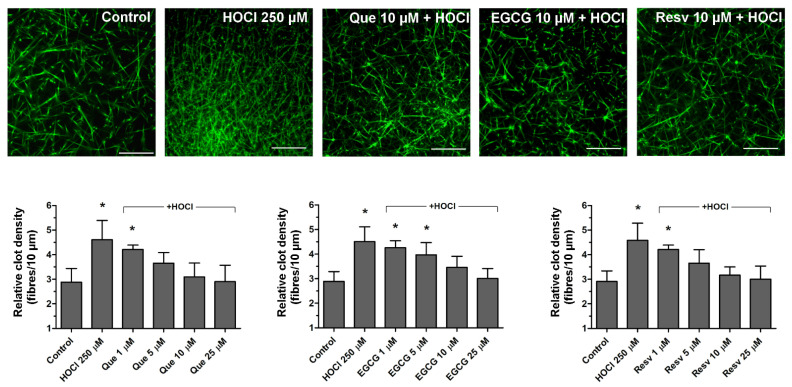
Effects of quercetin, epigallocatechin gallate, and resveratrol on HOCl-related increase in fibrin clot density. Plasma samples were incubated with appropriate vehiculum (ethanol or water) or with HOCl (250 μM for 5 min) or with indicated concentrations of Que, EGCG, or Resv, added 1 min before HOCl. Afterward, samples were supplemented with AF488-fibrinogen (to visualize fibrin formation) and clotting was triggered by recalcification (20 mM CaCl_2_, final conc.). Result fibrin clots were analyzed under confocal microscope toward fibrin density. Structures representative of six independent experiments are presented. Bars are means ± S.D. from six experiments. * *p* < 0.05 vs. control. Distance bar is 30 μm.

**Figure 4 antioxidants-11-00779-f004:**
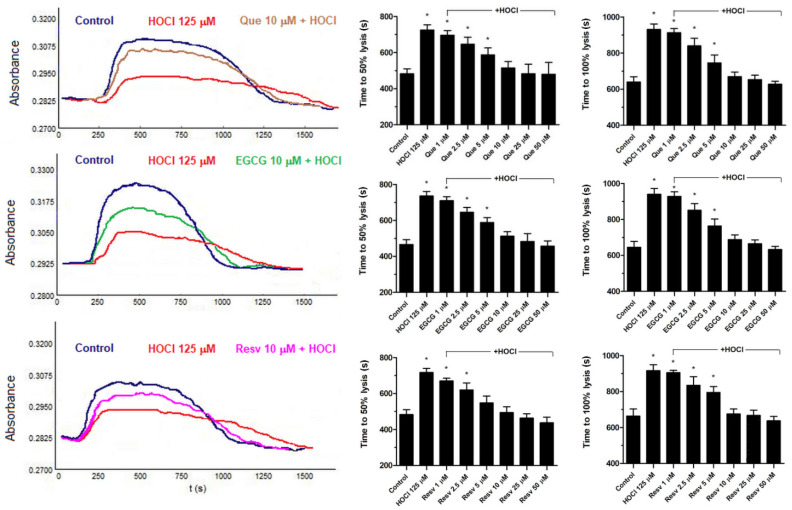
Effects of quercetin, epigallocatechin gallate, and resveratrol on HOCl-evoked attenuation of fibrinolysis. Plasma samples were incubated with appropriate vehiculum (ethanol or water) or with HOCl (125 μM for 5 min) or with indicated concentrations of Que, EGCG, or Resv, added 1 min before HOCl. Coagulation was triggered by the addition of CaCl_2_ (to 20 mM final concentration). The time to 50% and 100% of lysis was determined as the time required for a decrease in absorbance (of 50% or 100%, respectively) after reaching the maximal turbidity, measured in the presence of tissue plasminogen activator (200 ng/mL final concentration). Representative lysis profiles from six experiments are presented. * *p* < 0.05 vs. control.

**Figure 5 antioxidants-11-00779-f005:**
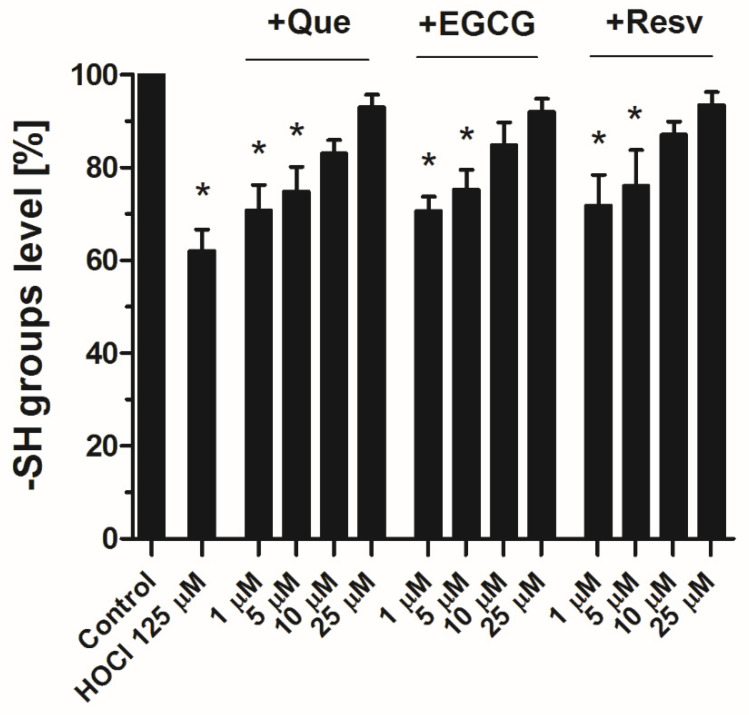
Effects of quercetin, epigallocatechin gallate and resveratrol on HOCl-produced decrease in free sulfhydryl groups in plasma. Samples of plasma were incubated for 10 min at 37 °C with HOCl (125 μM) without (control) or with, preincubated or not with studied antioxidants (for 1 min) at indicated concentrations. The content of reduced thiol (-SH) groups in plasma was measured using the DTNB assay. The total free-SH group content in plasma was about 533.5 ± 83 μM. Data are mean values ± S.D. from three independent experiments (each in triplicate). * *p* < 0.05.

## Data Availability

All of the data is contained within the article.

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
