# Peer review of "Natural Polyphenols May Normalize Hypochlorous Acid-Evoked Hemostatic Abnormalities in Human Blood"

_antioxidants, 2022, doi:10.3390/antiox11040779_

Round 1

Reviewer 1 Report

The present manuscript was aimed to investigate three plant-derived polyphenols with well-known antioxidant properties at concentration not affecting platelet responses per se, may normalize particular aspects of hemostasis disturbed by HOCl. The results indicated that supplementation with quercetin and epigallocatechin gallate may be helpful in normalization of hemostatic abnormalities during inflammatory states associated with elevated HOCl production, while presence of resveratrol enhanced the inhibitory action of HOCl towards platelets. These experimental data were interesting, however, the present results only displayed preliminary advances and no significant improvements on the bioactive components discovery related fields could be observed. In summary, this manuscript is not recommended to accept for publication in Antioxidants. In addition, there are several major comments to be addressed as following.

  1. According to the results in this study, the window of higher concentration and lower concentration was narrow. It limited the future development of these compounds.
  2. The mechanism of these drugs was not studied. Only discussion was not enough to evidence the possible mechanism.
  3. There were not any in vivo data in this manuscript. In addition, the reasonable concentrations of these compounds were difficult to be reached in in vivo model.
  4. In the References section, the writing manner of several references did not follow the style of this journal. Authors have to check and revise these errors, including refs 3, 7, 12, 13, 20, 27, 39, 46, and 51.

Reviewer 2 Report

This study investigate effects of natural polyphenols on clot formation and if they may normalize some aspects of hemostasis disturbed by HOCl due to their antioxidative properties. The design of the study, methodology, results and interpretation of results are sound and conclusions are drawn from the results.

Have you assessed any of antioxidative parameters, since this effect  (antioxidative properties of natural polyphenols and effect on HOCl) is in the hypothesis of the study? Would you consider an additional control experiments with measurements of oxidative stress, or at least provide some theoretical coverage for the mechanisms of observed effects? It would add a value to mechanistic aspect of the study, since at this level, it does not explain the mechanisms, but provide description of the effects and interactions between HOCl and particular polyphenol that was used.

I appreciate limitation of the study in terms of clinical application by dietary intake of polyphenols. In that term, what would be the ultimate meaning of the results?

Author Response

Antioxidants Rebuttal                          ref 1638825

Misztal et al.

Antioxidants

ref 1638825

4th April, 2022

We thank the Editors for the opportunity to submit our manuscript “Natural polyphenols may normalize hypochlorous acid-evoked hemostatic abnormalities in human blood” to Antioxidants for consideration for publication.

We provide a comprehensive rebuttal (below, responses in red italics) to the Reviewers’ comments. We have revised our manuscript accordingly and have provided an amended version with tracked changes as requested. We also conducted additional experiments (provided as Figure 5).

We once again thank the Editors of Antioxidants for considering our manuscript for publication.

With kind regards,
Dr Tomasz Misztal

Reviewer 2

This study investigate effects of natural polyphenols on clot formation and if they may normalize some aspects of hemostasis disturbed by HOCl due to their antioxidative properties. The design of the study, methodology, results and interpretation of results are sound and conclusions are drawn from the results.

Have you assessed any of antioxidative parameters, since this effect (antioxidative properties of natural polyphenols and effect on HOCl) is in the hypothesis of the study? Would you consider an additional control experiments with measurements of oxidative stress, or at least provide some theoretical coverage for the mechanisms of observed effects? It would add a value to mechanistic aspect of the study, since at this level, it does not explain the mechanisms, but provide description of the effects and interactions between HOCl and particular polyphenol that was used.

Response:

We thank the Reviewer for raising important issues. Guided by suggestions from the Reviewer, we conducted additional experiment to quantify reduced -SH groups in plasma samples – enriched or not with studied antioxidants – to better evaluate antioxidative-based mechanism of normalized hemostasis in plasma exposed to HOCl (please see Fig. 5 in revised manuscript).

I appreciate limitation of the study in terms of clinical application by dietary intake of polyphenols. In that term, what would be the ultimate meaning of the results?

Response:

We thank the Reviewer for this comment and for the opportunity to clarify this issue. We are conscious that our approach eliminated lower concentrations of HOCl (which also can affect hemostasis. Please see Misztal et al., 2019) and, correspondingly, even lower concentrations of antioxidants, more easily achieved in vivo (especially via novel supplementation strategies mentioned in text). We added the following sentence in the Discussion section (in “limitation” part):

“It has also be stressed that our approach considered specific range of the stressor concentrations, i.e. those producing about 50% of inhibition/augmentation (dependent on specific parameter). One can expect that lower concentrations of HOCl, also producing hemostatic abnormalities (and more likely to occur in blood during inflammation [15]), would be neutralized by even lower – and hence more easily achieved in vivo – concentrations of specific antioxidants.” 

  1. Misztal et al. The myeloperoxidase product, hypochlorous acid, reduces thrombus formation under flow and attenuates clot retraction and fibrinolysis in human blood. Free Radic Biol Med. 2019, 141, 426-437.

Reviewer 3 Report

I’ve read with attention the paper of Misztal et al. that is potentially of interest. The background and aim of the study have been clearly defined. The methodology applied is described in detail and overall correct, the results are reliable and adequately discussed. The only paragraph that should be modified is the following one "Since aspirin is commonly used to reduce symptoms of inflammation (beyond its antiplatelet action), and polyphenols are increasingly recognized as beneficial antioxidants, the simultaneous exposure of platelets to aspirin and Resv during conditions associated with elevated HOCl production is probable, which may potentially result in unexpected bleeding tendencies. Therefore, there is an urgent need to explore interactions between polyphenols and classical antiplatelet drugs more deeply, considering broader concentrations of polyphenols (not only within antiplatelet range). Our research can make a significant contribution to this field as a preliminary study." that is speculative and not supported by any reference. 

Reviewer 4 Report

This article  presents the data concerning hypochlorous acid (HOCl) on platelet function.  This is very important problem because during pathogen invasion neutrophils generate  large amount of HOCl.  Inflammation may lead to overproduction of HOCl and induce modification plasma proteins involved in hemostasis and platelet functions. In the work polyphenolic compounds were used: quercetin, epigallocatechin gallate (EGCG) and resveratrol. It has been shown that the polyphenols differently regulate platelet aggregation (at higher concentrations) and hemostasis / fibrinolysis (at low concentrations) in the presence of HOCl. The report indicates that quercetin and EGCG normalize hemostatic abnormalities during elevated HOCl production in contrast to resveratrol which enhanced inhibitory action of HOCL toward platelets. The authors explain the inhibitory action of resveratrol  by activation of the platelet inhibitory pathway (NO/cGMP/PKG). However, it has been shown also quercetin and EGCG activate another platelet inhibitory pathway (cAMP/PKA). All three compounds decrease calcium influx which affects platelet activation.  In the article Oh _WJ et al 2012, for example quercetin at relatively low concentrations induced very strong inhibition of the Ca2+ influx. The conclusion of the authors that ”the antioxidative properties  are the primary mechanism of Que and EGCG at this concentration range (below antiplatelet range” is not  clearly proved. The experiments are well-designed and done, however the interpretation of data in the discussion are not sufficient. Inhibition of platelet aggregation by HOCl might be mediated by different mechanisms including direct effect of HOCl on platelet activatory (Calcium mobilization, direct inactivation of collagen receptors, etc), or inhibitory (activation of cyclic nucleotides system) pathways. Aggregation also will be inhibited when platelets become procoagulant, or undergo to apoptosis. Data on mechanisms of HOCl induced inhibition of platelet aggregation are important for understanding of more detailed mechanisms of its prevention by used compounds. Another important question connected with the use of collagen to induce platelet aggregation. Will be good to use other agonists (ADP, TRAP, and others), such data also could help for interpretation of the results. If not considering additional experiments, the authors should include these questions in the discussion    

Author Response

Antioxidants Rebuttal                          ref 1638825

Misztal et al.

Antioxidants

ref 1638825

4th April, 2022

We thank the Editors for the opportunity to submit our manuscript “Natural polyphenols may normalize hypochlorous acid-evoked hemostatic abnormalities in human blood” to Antioxidants for consideration for publication.

We provide a comprehensive rebuttal (below, responses in red italics) to the Reviewers’ comments. We have revised our manuscript accordingly and have provided an amended version with tracked changes as requested. We also conducted additional experiments (provided as Figure 5).

We once again thank the Editors of Antioxidants for considering our manuscript for publication.

With kind regards,
Dr Tomasz Misztal

Reviewer 4

This article  presents the data concerning hypochlorous acid (HOCl) on platelet function.  This is very important problem because during pathogen invasion neutrophils generate  large amount of HOCl.  Inflammation may lead to overproduction of HOCl and induce modification plasma proteins involved in hemostasis and platelet functions. In the work polyphenolic compounds were used: quercetin, epigallocatechin gallate (EGCG) and resveratrol. It has been shown that the polyphenols differently regulate platelet aggregation (at higher concentrations) and hemostasis / fibrinolysis (at low concentrations) in the presence of HOCl. The report indicates that quercetin and EGCG normalize hemostatic abnormalities during elevated HOCl production in contrast to resveratrol which enhanced inhibitory action of HOCL toward platelets. The authors explain the inhibitory action of resveratrol  by activation of the platelet inhibitory pathway (NO/cGMP/PKG). However, it has been shown also quercetin and EGCG activate another platelet inhibitory pathway (cAMP/PKA). All three compounds decrease calcium influx which affects platelet activation.  In the article Oh _WJ et al 2012, for example quercetin at relatively low concentrations induced very strong inhibition of the Ca2+ influx. The conclusion of the authors that ”the antioxidative properties  are the primary mechanism of Que and EGCG at this concentration range (below antiplatelet range” is not  clearly proved. The experiments are well-designed and done, however the interpretation of data in the discussion are not sufficient. Inhibition of platelet aggregation by HOCl might be mediated by different mechanisms including direct effect of HOCl on platelet activatory (Calcium mobilization, direct inactivation of collagen receptors, etc), or inhibitory (activation of cyclic nucleotides system) pathways. Aggregation also will be inhibited when platelets become procoagulant, or undergo to apoptosis. Data on mechanisms of HOCl induced inhibition of platelet aggregation are important for understanding of more detailed mechanisms of its prevention by used compounds. Another important question connected with the use of collagen to induce platelet aggregation. Will be good to use other agonists (ADP, TRAP, and others), such data also could help for interpretation of the results. If not considering additional experiments, the authors should include these questions in the discussion.    

Response:

We thank the Reviewer for this comment and for the opportunity to clarify this issues. Please see our response below:

  1. However, it has been shown also quercetin and EGCG activate another platelet inhibitory pathway (cAMP/PKA). All three compounds decrease calcium influx which affects platelet activation.  In the article Oh _WJ et al 2012, for example quercetin at relatively low concentrations induced very strong inhibition of the Ca2+ influx.

Response:

One problem with the measuring of calcium influx into activated platelets is that most publications refer to fluorimetric method conducted using washed platelets suspension (platelet suspended in artificial medium). Such approach is however, strongly non physiological and hence hard to extrapolate on in vivo situation (platelets in whole blood). For example, other naturally occurred, inflammation-related strong oxidant, peroxynitrite may exert paradoxical effects on platelet, dependent whether it is studied in washed platelet model (where it induces platelet aggregation [Brown et al., 1998]) or in platelet-rich plasma (where it inhibits platelet responses [Rusak et al., 2006]). Similarly, our previous observations suggest that HOCl, at low micromolar concentrations reduces thrombin-evoked calcium influx to platelets dose-dependently, while higher concentrations of stressor induces calcium signal in platelet cytosol which was directly associated with the damage of platelet plasma membrane (please compare Fig. 6 and Tab. I from Misztal et al., 2014). To resolve the above problem, whole blood assay with confocal microscopy and assessment of calcium signal in single platelets under flow conditions after activation would be preferable [Agbani et al., 2015]. Unfortunately, we do not have eventuality to perform such experiments in this time.

A.S. Brown, M.A. Moro, J.M. Masse, E.M. Cramer, M. Radomski, V. Darley-Usmar. Nitric oxide-dependent and independent effects on human platelets treated with peroxynitrite. Cardiovascular Research. 1998, 40(2), 380-388.

  1. RusakM. TomasiakM. Ciborowski. Peroxynitrite can affect platelet responses by inhibiting energy production. Acta Biochim Pol. 2006, 53(4), 769-776

  1. MisztalT. RusakM. Tomasiak. Clinically relevant HOCl concentrations reduce clot retraction rate via the inhibition of energy production in platelet mitochondria. Free Radic Res. 2014, 48(12), 1443-1453.

E.O. Agbani et al. Coordinated membrane ballooning and procoagulant spreading in human platelets. Circulation. 2015, 132(15), 1414-1424.

  1. The conclusion of the authors that ”the antioxidative properties  are the primary mechanism of Que and EGCG at this concentration range (below antiplatelet range” is not  clearly proved.

Response:

In line to better evidence the antioxidative properties of all three studied antioxidants as a primary anti-HOCl mechanism in plasma, we conducted additional experiments where we quantified reduced -SH groups in HOCl-exposed plasma samples – enriched or not with studied antioxidants (please see Figure 5 in revised manuscript).

  1. Inhibition of platelet aggregation by HOCl might be mediated by different mechanisms including direct effect of HOCl on platelet activatory (Calcium mobilization, direct inactivation of collagen receptors, etc), or inhibitory (activation of cyclic nucleotides system) pathways. Aggregation also will be inhibited when platelets become procoagulant, or undergo to apoptosis.

Response:

While we named inhibition of calcium influx as a potential mechanism of HOCl-reduced platelet aggregation (next to inhibition of mitochondrial respiration, lines 307-308), other mechanisms pointed by the Reviewer are potentially possible. However, since in our previous study we demonstrated that HOCl even at very high concentrations (up to 1 mM) is unable to evoke procoagulant platelet phenotype per se [Misztal et al.,2019, Figure 2C] we believe that this is not the case. Since the precise molecular mechanism of HOCl-associated platelet inhibition was not a goal of our study, we based the discussion of this subject on the available literature.

  1. Misztal et al. The myeloperoxidase product, hypochlorous acid, reduces thrombus formation under flow and attenuates clot retraction and fibrinolysis in human blood. Free Radic Biol Med. 2019, 141, 426-437.

  1. Another important question connected with the use of collagen to induce platelet aggregation. Will be good to use other agonists (ADP, TRAP, and others), such data also could help for interpretation of the results. If not considering additional experiments, the authors should include these questions in the discussion.

Response:

We used collagen for particular reason, i.e. collagen is initial platelet agonist during arterial thrombosis (which is more platelet-dependent compared to venous thrombosis [Warny et al., 2019; Freedman, 2005]) which is mostly associated with damage of a vessel wall and subsequent exposure of collagen in vessel lumen. Such scenario (damage of vessel wall) is often associated with pathogen invasion and subsequent neutrophils activation and HOCl production. What is more, activation of platelet by collagen is intrinsically connected with the generation of reactive oxygen species, necessary to enhance the initial activation cascade (Qiao et al., 2018; Carrim et al., 20014; Jang et al., 2014). Considering above, we believe that using collagen in a study comprising reactive species (HOCl) and antioxidants is rational.

  1. Warny et al. Arterial and venous thrombosis by high platelet count and high hematocrit: 108 521 individuals from the Copenhagen General Population Study. J Thromb Haemost. 2019, 17(11), 1898-1911.

J.E. Freedman. Molecular regulation of platelet-dependent thrombosis. Circulation. 2005, 112(17), 2725-3427.

  1. Carrim et al. Role of focal adhesion tyrosine kinases in GPVI-dependent platelet activation and reactive oxygen species formation. PLoS One. 2014, 9(11), e113679.

J. Qiao et al. Regulation of platelet activation and thrombus formation by reactive oxygen species Redox Biology. 2018, 14, 126-130.

J.Y. Jang et al. Reactive Oxygen Species Play a Critical Role in Collagen-Induced Platelet Activation via SHP-2 Oxidation. Antioxid Redox Signal. 2014, 20(16): 2528–2540.

Other platelets stimuli are tempting to study in future since it was not a primary goal in our present investigation. Initially, it seems that HOCl is a relatively weak inhibitor of ADP-evoked platelet aggregation (in study of Murine et al. HOCl exerted strong effect at 0.15 mM concentration but in a washed platelets suspension). On the other hand, study of our team suggests that HOCl is able to inhibit platelet secretion (measured as P-selectin exposure) in TRAP-activated platelets (in plasma) at high concentration (0.5 mM) while it was unable to diminish procoagulant phosphatidylserine exposure in collagen-stimulated platelets at concentrations up to 1 mM [Misztal et al., 2019].

M.A. MurinaE.L. Savel'evaD.I. Roshchupkin. Mechanism of action of biogenic chloramines and hypochlorite on initial aggregation of blood platelets. Biofizika. 2006, 51(2), 299-305.

  1. Misztal et al. The myeloperoxidase product, hypochlorous acid, reduces thrombus formation under flow and attenuates clot retraction and fibrinolysis in human blood. Free Radic Biol Med. 2019, 141, 426-437.

“We introduced the following fragment in the Discussion section: “Since in this report platelet aggregation and thrombus formation was triggered solely by collagen the question arises what might be the impact of HOCl and antioxidants on platelet responses evoked by other platelet stimuli, e.g. ADP or thrombin receptor agonists? HOCl seems to be rather a weak inhibitor of thrombin receptor activating peptide (TRAP)-induced platelet activation since significant reduction of platelet secretion (evaluated as P-selectin exposure) triggered ty this agonist was observed at relatively high (≥0.5 mM) concentration of stressor [15]. On the other hand, 0.15 mM HOCl strongly inhibited platelet aggregation induced by ADP, in a washed platelet suspension [67]. Potential impact of antioxidants on platelet responses, triggered by different agonist than collagen, in the presence of HOCl is a valid subject for future investigations.

Reviewer 5 Report

The authors presented work demonstrating that well-described polyphenols can modulate haemostatic function in the presence of HOCl - a known inducer of abnormal haemostasis. The manuscript is very well presented, written and the experiments are well described. Overall, an excellent manuscript difficult to critique.

My only comment is how the authors chose 500 uM HOCl as the pathophysiological range in the setting of abnormal haemostasis? An explanation in the manuscript would be useful to readers. Could abnormal haemostasis be achieved with less HOCl, say 50 uM? If so, then you might see an affect with even lower concentrations of polyphenols that are more likely to be present in the serum (uM range is excessive for serum polyphenol levels). This would represent efficacy at physiological ranges. This should at least be discussed. 

Author Response

Antioxidants Rebuttal                          ref 1638825

Misztal et al.

Antioxidants

ref 1638825

4th April, 2022

We thank the Editors for the opportunity to submit our manuscript “Natural polyphenols may normalize hypochlorous acid-evoked hemostatic abnormalities in human blood” to Antioxidants for consideration for publication.

We provide a comprehensive rebuttal (below, responses in red italics) to the Reviewers’ comments. We have revised our manuscript accordingly and have provided an amended version with tracked changes as requested. We also conducted additional experiments (provided as Figure 5).

We once again thank the Editors of Antioxidants for considering our manuscript for publication.

With kind regards,
Dr Tomasz Misztal

Reviewer 5

The authors presented work demonstrating that well-described polyphenols can modulate haemostatic function in the presence of HOCl - a known inducer of abnormal haemostasis. The manuscript is very well presented, written and the experiments are well described. Overall, an excellent manuscript difficult to critique.

My only comment is how the authors chose 500 uM HOCl as the pathophysiological range in the setting of abnormal haemostasis? An explanation in the manuscript would be useful to readers. Could abnormal haemostasis be achieved with less HOCl, say 50 uM? If so, then you might see an affect with even lower concentrations of polyphenols that are more likely to be present in the serum (uM range is excessive for serum polyphenol levels). This would represent efficacy at physiological ranges. This should at least be discussed.

Response:

We thank the Reviewer for this restorative comment and for the opportunity to clarify the raised issue. We selected HOCl concentrations during preliminary experiments, in way to find specific concentrations producing ~ 50% of inhibition/enhancement (regarding the measured parameter) and then we applied those concentrations of stressor during particular research tasks. 500 μM was the highest concentrations of HOCl used, however one may expect even higher ones during severe inflammation (line 46 in Introduction). Like the Reviewer pointed, in most cases however, expected HOCl concentrations would be lower but still able to exert effect on haemostasis (please see reference [15] for threshold concentrations of HOCl in relation to different haemostatic responses). In regard to the above issue we propose the following fragment in the Discussion section (“study limitation” part):

“It has also to be stressed that our approach considered specific range of the stressor concentrations, i.e. those producing about 50% of inhibition/augmentation (dependent on specific parameter). One can expect that lower concentrations of HOCl, also producing hemostatic abnormalities (and more likely to occur in blood during inflammation [15]), would be neutralized by even lower – and hence more easily achieved in vivo – concentrations of specific antioxidants.” 

Round 2

Reviewer 1 Report

I had gone through the revised manuscript and author had provided some rational responses towards my previous queries. I still stand on my previous concerns that future developments of these polyphenols were limited and authors have to mention this clearly in the manuscript.  Although I was not fully satisfied with the answers provided by authors, this manuscript could be recommended to accept for publication in Antioxidants after some minor revision. Since there were no in vivo data and known mechanism, I would suggest this study to be accepted as a note.

Author Response

Antioxidants Rebuttal (2nd round)                     ref 1638825

Misztal et al.

Antioxidants

ref 1638825

6th April, 2022

We would like to thank the Reviewer for the opportunity to improve manuscript during 2nd round of the review process. We provide a comprehensive rebuttal (below, responses in red italics) to the Reviewers’ comments. We have revised our manuscript accordingly and have provided an amended version with tracked changes as requested.

Please find our response below.

Reviewer 1

I had gone through the revised manuscript and author had provided some rational responses towards my previous queries. I still stand on my previous concerns that future developments of these polyphenols were limited and authors have to mention this clearly in the manuscript.  Although I was not fully satisfied with the answers provided by authors, this manuscript could be recommended to accept for publication in Antioxidants after some minor revision. Since there were no in vivo data and known mechanism, I would suggest this study to be accepted as a note.

Response:

In line to better underline the future potential of studied compounds we propose the following paragraph in the Conclusion section:

“Coupling of the monitored supplementation of certain polyphenols using novel delivery strategies with a clinical practice might prospectively serve as a new potential strategy to normalize hemostatic abnormalities during massive HOCl production (inflammatory states). Potentially, other reactive species with evidenced disruptive action on hemostasis, e.g. peroxynitrite [37,68,69], might be also neutralized by certain antioxidants. However, as in case of HOCl, the precise mechanism of such interactions should be elucidated, next to the more complex studies including in vivo models.”

  1. T. MisztalT. RusakM. Tomasiak. Peroxynitrite may affect clot retraction in human blood through the inhibition of platelet mitochondrial energy production. Thromb Res. 2014, 133(3), 402-411. https://doi.org/10.1016/j.thromres.2013.12.016.
  2. P. Pacher J.S. BeckmanL. Liaudet. Nitric oxide and peroxynitrite in health and disease. Physiol Rev. 2007, 87(1), 315-424. https://doi.org/10.1152/physrev.00029.2006.

We fully agree with the Reviewer that in vivo study is a tempting idea and this is certainly something we look to doing in future larger studies. Unfortunately, we are unable at the moment to conduct additional examinations due to funding constraints.

In the matter of potential mechanism, we conducted a preliminary examination (n=2) of static platelet adhesion to collagen. The essence of this assay relies on relatively long (30-60 min) contact of platelet suspension with collagen-coated well of a microtiter plate and as such is sensitive to a direct impairment of collagen receptors. Since we did not notice any changes in static adhesion in the presence of HOCl (up to 1 mM), we believe that direct distortion of collagen receptors by this stressor is rather neglible.

However, since the small number of samples were analyzed, we decided to do not introduce this information in the present manuscript. This is, however, a good starting point for future investigation focused on the exact mechanism of HOCl/polyphenols action toward platelets.

According to the Reviewer’s suggestion, we are open to change the type of article (beyond full Article) if the Editor will find this clue suitable.

With kind regards,
Dr Tomasz Misztal